# Pinhole-seeded lateral epitaxy and exfoliation of GaSb films on graphene-terminated surfaces

Sebastian Manzo[1], Patrick J. Strohbeen[1], Zheng Hui Lim[1], Vivek Saraswat[1], Dongxue Du[1], Shining Xu[2], Nikhil Pokharel[2], Luke J. Mawst[2], Michael S. Arnold [1] & Jason K. Kawasaki [1✉]

Remote epitaxy is a promising approach for synthesizing exfoliatable crystalline membranes and enabling epitaxy of materials with large lattice mismatch. However, the atomic scale mechanisms for remote epitaxy remain unclear. Here we experimentally demonstrate that GaSb films grow on graphene-terminated GaSb (001) via a seeded lateral epitaxy mechanism, in which pinhole defects in the graphene serve as selective nucleation sites, followed by lateral epitaxy and coalescence into a continuous film. Remote interactions are not necessary in order to explain the growth. Importantly, the small size of the pinholes permits exfoliation of continuous, free-standing GaSb membranes. Due to the chemical similarity between GaSb and other III-V materials, we anticipate this mechanism to apply more generally to other materials. By combining molecular beam epitaxy with in-situ electron diffraction and photoemission, plus ex-situ atomic force microscopy and Raman spectroscopy, we track the graphene defect generation and GaSb growth evolution a few monolayers at a time. Our results show that the controlled introduction of nanoscale openings in graphene provides an alternative route towards tuning the growth and properties of 3D epitaxial films and membranes on 2D material masks.

[1] Materials Science and Engineering, University of Wisconsin-Madison, Madison, WI 53706, USA. [2] Electrical and Computer Engineering, University of Wisconsin-Madison, Madison, WI 53706, USA. ✉email: jkawasaki@wisc.edu

Remote epitaxy of three-dimensional materials on graphene-terminated surfaces[1] promises to circumvent many of the limitations of conventional epitaxy. For example, the weak van der Waals interaction between the film and graphene creates new strain relaxation pathways for highly mismatched growth[2,3] and enables exfoliation of free-standing crystalline membranes for flexible devices[1,4] and tuning properties via strain in novel geometries[5]. Since the first demonstrations for compound semiconductors[1,6], epitaxy and exfoliation from graphene have been demonstrated for a variety of other materials including transition metal oxides[4,7], simple metals[8], halide perovskites[3], and intermetallic compounds[5]. These films are typically thought to grow via a remote epitaxy mechanism[1,6], in which epitaxial registry between film and substrate is achieved via remote interactions that permeate through graphene.

The atomic scale mechanisms of remote epitaxy, however, remain unclear. Model calculations suggest that for perfect graphene/substrate interfaces, the substrate lattice potential permeates through graphene and may be sufficient to template epitaxial growth[1,3,6,8]. However, the magnitude of the potential fluctuations is only $\Delta\phi = 0$–40 meV, which is smaller than the $k_B T \sim 70$ meV thermal energy at typical growth temperatures 500 °C. Experimentally, the ability to grow and exfoliate crystalline films from graphene is often cited as evidence for growth via these remote (screened) interactions[1,3,7]. In practice, however, it remains an outstanding challenge to fabricate defect-free graphene or perfect graphene/substrate interfaces, especially because graphene cannot be grown directly on arbitrary substrates. Additional layer transfer steps are often required. The effects of native and transfer-induced defects in graphene, which are difficult to control[9–12], are generally overlooked in the remote epitaxy mechanism. Microscopic measurements during growth are required to understand the atomic-scale mechanisms for growth under realistic conditions.

Here we experimentally demonstrate that pinhole-seeded lateral epitaxy explains the growth of atomically smooth, exfoliatable GaSb films on a graphene-terminated GaSb (001) substrate. Using the same graphene transfer procedures as previous reports[1,3,13], we find that pinholes are created by native oxide desorption from the substrate. GaSb nucleates directly from the underlying GaSb substrate in the pinholes, followed by lateral overgrowth and coalescence of a continuous film. The resulting films have similar structural quality as previous reports[1–4,6,8] and can be exfoliated to produce free-standing membranes. Since GaSb shares similar bonding character, constituent vapor pressures, surface diffusion coefficients, and growth conditions as other III-V materials, we anticipate that the seeded lateral epitaxy mechanism may explain the growth of other materials on transferred graphene. We experimentally track the evolution of GaSb films grown on graphene/GaSb(001) by combining molecular beam epitaxy (MBE) synthesis with in-situ electron diffraction and photoemission spectroscopy (XPS), plus ex-situ atomic force microscopy (AFM) and scanning electron microscopy (SEM).

This growth mode is conceptually similar to selective area epitaxy and epitaxial lateral overgrowth (ELO), that employ openings in a dielectric mask such as $SiO_2$ to seed growth[14–18]. It improves upon conventional ELO in two key ways. First, due to the weak Van der Waals interactions and small pinholes (10–300 nm diameter), graphene masks permit the etch-free exfoliation of large scale single-crystalline GaSb membranes for applications in flexible electronics. Second, the atomic thinness of graphene is an attractive feature compared to conventional ELO masks, for applications where an electronically transparent interface is desired, e.g. a tunnel junction[19,20]. Seeded lateral epitaxy also carries an advantage over purely remote epitaxy: whereas remote epitaxy is only expected to apply for substrates with polar bonding[6], seeded lateral epitaxy can produce epitaxial films on nonpolar substrates[15]. Remote epitaxy also requires clean graphene/substrate interfaces, free from interfacial oxides or other contaminants[1]. Our experiments demonstrate that the introduction of defects provides powerful knob for controlling epitaxial growth of covalent 3D materials on 2D materials surfaces, similar to recent work on growth of 2D materials on patterned 2D masks[21,22].

## Results

**Epitaxy and exfoliation of GaSb films**. We first demonstrate the epitaxy and exfoliation of single crystalline GaSb films on a graphene-terminated GaSb (001) substrate. Figure 1(a, b) shows reflection high energy electron diffraction (RHEED) patterns tracking the MBE-growth of GaSb directly on GaSb (001) and on graphene-terminated GaSb (001). On this sample, we layer transfer graphene on half of the GaSb (001) substrate using a wet transfer procedure similar to the one that has previously been used for remote epitaxy of GaAs, including an HCl etch to remove native oxides from the substrate[1,13] (Methods). HCl[23–25] and HCl+$H_2O_2$ solutions[26,27] are also commonly used to etch native oxides from GaSb. The other half of the sample has an exposed GaSb surface for direct epitaxy.

Immediately prior to growth we follow the standard procedure of annealing at 520 °C in an $Sb_2/Sb_1$ flux to desorb the native oxides from the substrate. On the bare GaSb side, the native oxide removal is confirmed by the appearance of bright semi-streaky spots in the RHEED pattern. The graphene-terminated side looks qualitatively similar, although the pattern is not as bright and there is diffuse intensity around the specular reflection. We attribute these differences in part to scattering through monolayer graphene that is randomly oriented in-plane.

After 75 nm of GaSb film growth, on both sides of the sample we observe a distinct ring of spots RHEED pattern with $3\times$ superstructure reflections corresponding to the expected $c(2\times 6)$ reconstruction of GaSb (001) [Fig. 1(a, b), bottom]. The clear superstructure indicates that a well-ordered, smooth epitaxial film has grown on both the bare substrate side and on the graphene-terminated side. X-ray diffraction measurements confirm epitaxial GaSb growth with no impurity phases or orientations (Fig. 1d). Closer inspection of the 006 reflection reveals a subtle shift of the film reflection to lower angle compared to the substrate, or a larger out-of-plane lattice parameter. We speculate that this shift may arise from strain induced by the underlying graphene, e.g. due to the large thermal expansion mismatch between graphene and GaSb. Alternatively, the shift may be due to slight wrinkling of the GaSb film since it resting on a graphene interlayer. XRD rocking curves reveal a similar picture. In the rocking curves of the 004 reflection for a 120 nm GaSb film on graphene/GaSb (Supplementary Fig. 1), in addition to the expected peak at $\omega = 0$, we observe a second peak shifted by $\omega = -15$ arc seconds. The full width at half maxima (fwhm) of these two peaks are 10.44 and 12.59 arc seconds, respectively, compared to a width of 6.74 arc seconds for a homoepitaxial film. In addition, photoluminescence measurements reveal similar materials quality between films grown on graphene/GaSb (001) and a bare GaSb substrate (Supplementary Fig. 2).

Our GaSb films can be mechanically exfoliated from the graphene-terminated side of the sample, using a Ni stressor layer (Methods). Figure 1(e, f) shows scanning electron micrographs of the top side of the substrate and the bottom side of the exfoliated membrane, after exfoliation. The large scale SEM images are comparable to previous demonstrations of remote epitaxy of other III-V materials, including GaAs and InP[1,6], and show minimal spalling marks or tears. $\omega - 2\theta$ X-ray diffraction

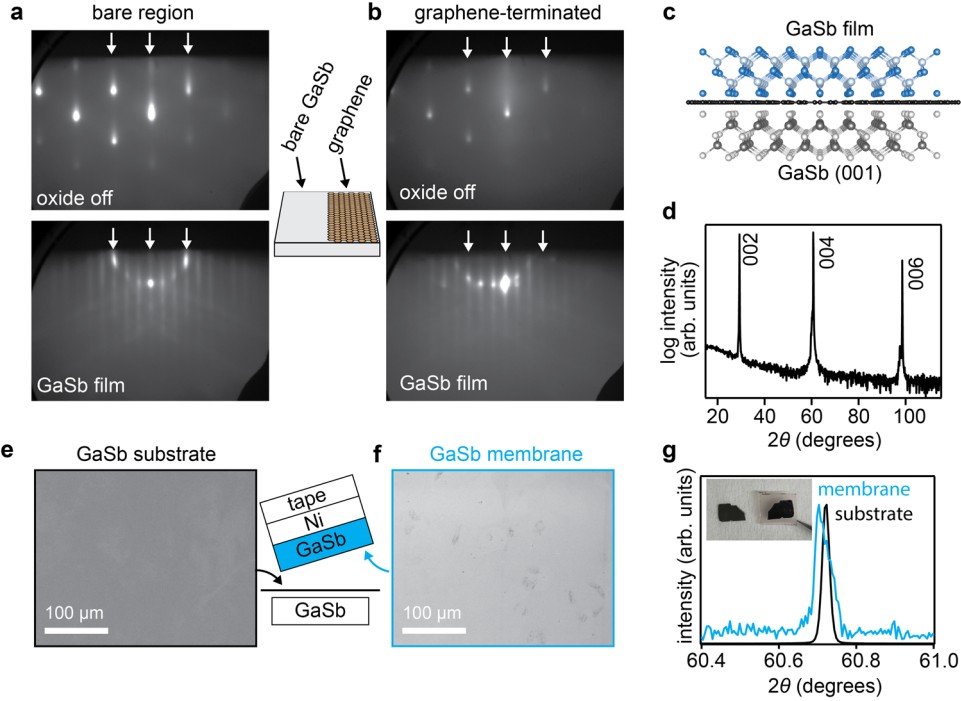

**Fig. 1 Epitaxy and exfoliation of GaSb on graphene-terminated GaSb (001). a–d** GaSb epitaxy on a GaSb (001) substrate that is half covered with graphene. **a** Reflection high energy electron diffraction (RHEED) patterns along a [1$\bar{1}$0] azimuth, tracking direct epitaxy on the exposed GaSb region. RHEED after the oxide off step shows a bright three-dimensional pattern. After 75 nm of film growth, an atomically smooth smooth ring of spots RHEED pattern is recovered. **b** RHEED patterns tracking oxide desorption and growth on the graphene-terminated half of the substrate. **c** Schematic of the heterostructure. (d) X-ray diffraction pattern of the GaSb film grown on the graphene-terminated side of GaSb (001). **e–g** GaSb membrane exfoliation using a Ni stressor layer. **e** SEM image of the under side of the exfoliated GaSb membrane. The inset schematic illustrates the exfoliation. **f** SEM images of the substrate after exfoliation. **g** X-ray diffraction patterns of the 004 reflection for the exfoliated GaSb membrane (blue) and the GaSb substrate (black). Insert: example photo of the substrate (left) and GaSb membrane (right) after exfoliation. Membrane dimensions are approximately 10 mm by 6 mm.

measurements of the exfoliated membranes (Fig. 1g, blue curve) confirm that the membrane itself is single crystalline. We find that the 004 reflection of the membrane is slightly shifted the lower angle and slightly asymmetric compared to the substrate 004 reflection. We attribute these differences to strain, ripples, or slight mis-alignments when measuring thin (75 nm) exfoliated membranes.

**Pinholes created by native oxide desorption**. We now show that GaSb films grow on graphene-terminated GaSb (001) via a lateral epitaxy, seeded at pinhole defects. We first examine the creation of pinhole defects in graphene, which form due to native oxide desorption from the substrate immediately prior to GaSb film growth.

Figure 2b shows two models of the graphene/GaSb substrate interface, immediately after the graphene layer transfer but before heating the sample to the film growth temperature. For an idealized transfer, the graphene/GaSb interface would be chemically clean and atomically sharp. This is a requirement for the substrate lattice potential to permeate through graphene and template film growth in a remote epitaxy mode[1]. However, such clean interfaces can be difficult to fabricate in practice, especially for III-V semiconductor substrates like GaSb since III-Vs are typically terminated with an amorphous ~3 nm thick native oxide[28]. This oxide must be removed before film growth in order to reveal the underlying crystalline template for an epitaxial film. Although we adopt an HCl etching procedure to remove the substrate native oxides prior to forming the graphene/substrate interface (Methods), some residual oxides may remain.

Our in-situ photoemission spectroscopy measurements reveal that some interfacial oxides remain after the etch procedure. Figure 2(a) shows in-situ x-ray photoemission spectra (XPS) of

the O 1s and Sb 3d core levels for a graphene-terminated GaSb(001) sample, as a function of annealing. These measurements are performed in an interconnected MBE + XPS chamber, such that samples are annealed and measured without removing them from ultrahigh vacuum ($p < 5 \times 10^{-10}$ Torr). After a 350 °C ultrahigh vacuum anneal to clean surface adsorbates, we observe a significant O 1s intensity at binding energy of 531 eV. In addition, we observe a satellite to the Sb $3d_{3/2}$ core level at binding energy of 540 eV, indicative of Sb-oxides.

Further annealing of our graphene/GaSb(001) samples removes most of the oxides, despite the presence of the graphene overlayer. After annealing at 515 °C with an $Sb_2/Sb_1$ flux, no oxide components are detected in the Sb 3d or Ga 3p core levels, suggesting that the GaSb-oxides have fully desorbed. We also observe a finite O 1s core level intensity that decreases as a result of the 515 °C anneal, and decreases further after the 540 °C anneal. At this point the chemical state of the remaining oxygen above 515 °C is unclear, since no oxide components are detected in the Ga 3p, Sb 3d, or C 1s core levels, suggesting that neither the GaSb substrate nor the graphene itself is oxidized (Supplementary Fig. 3). This suggests that the remaining oxygen at high temperature is weakly coordinated to the graphene or GaSb, yet remains trapped in the near surface region of the sample. Further experiments are needed to understand the oxygen that remains after nominal GaSb-oxide desorption. We observe similar native oxides for graphene/GaAs (001) interfaces even after the HCl etch (Supplementary Fig. 4). Variations on the anneal, including annealing in a $H_2$/Ar forming gas, also revealed residual oxygen and Sb-oxides after light annealing (Supplementary Fig. 5).

The interfacial oxide desorption creates pinholes in the graphene. Figure 3 tracks the evolution of surface morphology

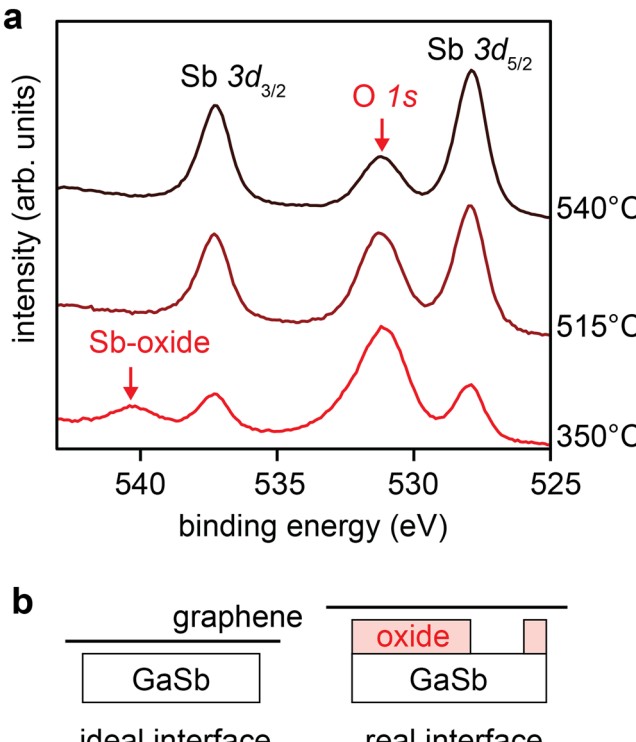

**Fig. 2 Native oxides desorption from the graphene/GaSb interface. a** In-situ photoemission spectra of the Sb 3*d* and O 1*s* core levels, tracking the desorption of native oxides by annealing. **b** Schematics of idealized and real interfaces, after a chemical etch but before thermal oxide desorption.

during native oxide desorption. The sample lightly annealed at 350 °C appears qualitatively smooth with no tears or holes observed in the SEM images (Fig. 3a). AFM topography reveals a bumpy morphology with height variations of ~2 nanometers and hints of the underlying GaSb step and terrace morphology, with terrace length ~100 nm (Fig. 3b, left). We attribute the slightly bumpy morphology to the presence of some interfacial oxides. Raman spectroscopy at this stage shows sharp *G* and 2*D* bands with no detectable *D* peak, which is commonly used as a metric to indicate minimal defects in the graphene. These measurements show that before oxide desorption, the transferred graphene has a relatively low defect density.

After annealing at 450 °C, the onset of native oxide desorption, pinholes with diameter ~10 nm are observed in the AFM phase and height images (Fig. 3b, c, e). The distinct contrast in the AFM phase suggests a different elastic modulus in the holes (exposed GaSb) compared to away from the pinholes (graphene). We interpret these holes to be created by the native oxide desorption process, as illustrated in Fig. 3(d). These pinholes are difficult to observe at SEM length scales (Fig. 3(a)) and are not readily detected by Raman spectroscopy since the sample does not have an observable *D* peak (Fig. 3f). The RHEED signature of the beginnings of oxide desorption is a sharpening of the specular reflection (Fig. 3a, middle insert). An extended analysis of the defects, strain, and doping extracted from Raman is shown in (Supplementary Fig. 6).

The pinholes appear to be localized at the bumps that were observed after annealing to 350 °C. However, a more detailed study of samples before and after de-oxidation, at exactly the same location, would provide a more definitive answer to their origin. Oxygen (and oxygen plasmas) are well known to etch

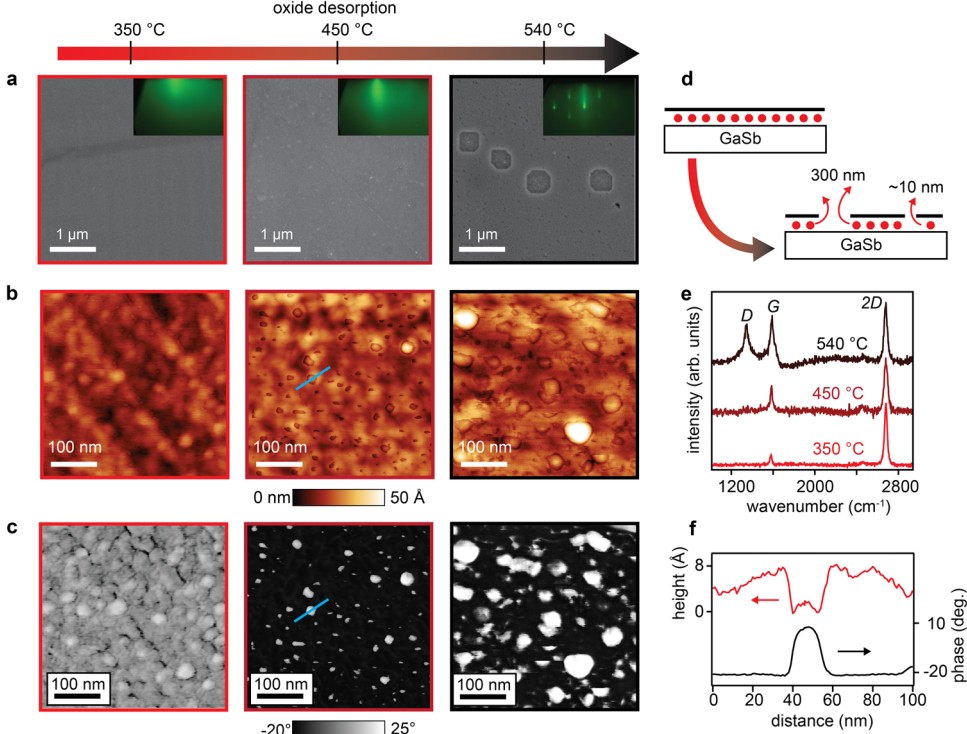

**Fig. 3 Pinholes created by native oxide desorption. a** SEM images and RHEED patterns tracking the native oxide desorption and creation of large 300 nm diameter holes. **b** AFM height images. Pinholes with diameter ~10 nm appear at the onset of native oxide desorption, 450 °C. **c** AFM phase contrast images. The phase contrast arises from differences in elastic modulus between the graphene and the holes with exposed GaSb. **d** Schematic of the oxide desorption process leading to the creation of holes with different sizes. Red circles represent oxides. **e** Raman spectra at various stages during oxide desorption. Strong *D* peak activation coincides with the appearance of large holes after annealing at 540 °C. **f** AFM height and phase line profiles extracted from the 450 °C sample. The corresponding line cut is marked by the blue line in the AFM images.

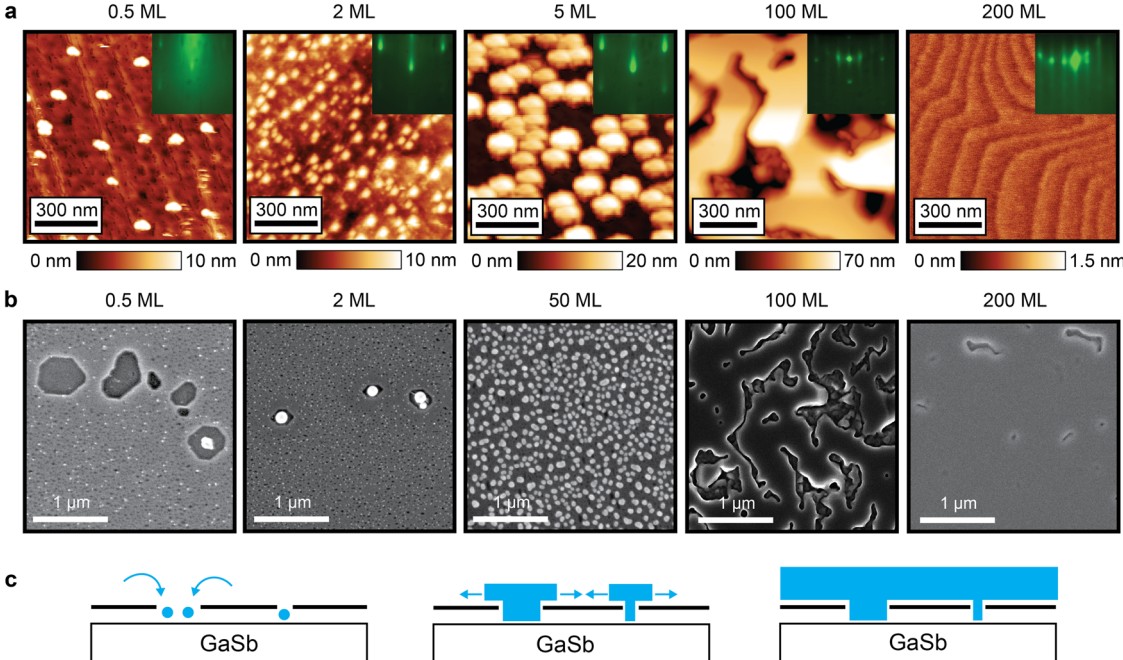

**Fig. 4 Seeded lateral epitaxy at the graphene pinholes. a** AFM images tracking the evolution of GaSb nucleation at pinholes in the graphene (<2 monolayers, ML), followed by lateral overgrowth (5 ML) and coalescence into an atomically stepped continuous film (200 ML). One monolayer is defined as half the thickness of a unit cell $a/2 = 3$ Angstrom, corresponding to an areal density of $5.4 \times 10^{14}$ atoms/cm$^2$ each of Ga and Sb. The GaSb growth temperature was 500 °C and the oxide desorption temperatures were 520−540 °C. Inserts show RHEED patterns at each stage of growth. **b** SEM images showing the nucleation at both small and large holes. The islands extend and coalesce around 50–100 ML, and nearly full coalescence is observed after 200 ML film growth. **c** Schematic of lateral epitaxy seeded at holes in the graphene. Blue circles represent GaSb nucleation.

graphene[29], particularly at edges and at extended defects like grain boundaries. In the present case, we expect that the larger strains around these protrusions may enhance the oxygen etch rate at these locations, compared to smoother regions of the graphene. Similar enhanced oxygen etching of graphene on other substrates has been attributed to roughness and impurities in the underlying substrate[30].

With further annealing up to 540 °C, the areal density and diameter of the pinholes increases as observed in the AFM images (Fig. 3b, c) and in Supplementary Figure 7. Additionally, we observe larger ~300 nm diameter holes in the SEM images. After the 540 °C anneal, a significant *D* band is present in the Raman spectrum, which is indicative of graphene defects. We observe similar pinhole formation for graphene on GaAs (001) (Supplementary Fig. 8).

An important question is whether the presence of oxides and creation of pinholes upon annealing are unique to our wet graphene transfer process, or if the creation of pinholes is more general. Note that we use the same HCl etching procedure that has been used previously for transferred graphene on III-Vs[1]. Additionally, recent XPS measurements show there are oxide satellites in the C 1*s* and As 3*d* core levels for both wet and dry transferred graphene on GaAs (001)[13]. Therefore, it appears that native oxides trapped at graphene/III-V interfaces are common to both wet (aqueous) and dry (in air)[31] transfer procedures, due to the reactivity of III-V surfaces[32]. Further improvements in interfacial cleanliness may require in-vacuum or glovebox transfer procedures[33].

**Seeded lateral epitaxy at pinholes**. These pinholes serve as nucleation sites for direct epitaxial growth. Figure 4 tracks the evolution of GaSb film growth on the graphene/GaSb (001) surface after native oxide desorption. After 0.5 monolayers (ML)

of GaSb growth, AFM and SEM measurements show that the GaSb islands selectively nucleate in the ~10 nm and 300 nm pinholes rather than on the graphene (Fig. 4a, b). We attribute the selective nucleation in the pinholes to the fact that exposed holes to the substrate are more chemically reactive than graphene. At this point we primarily observe nucleation on top of exposed regions of the GaSb substrate, rather than at graphene edges or underneath the graphene. Further experiments are required in order to comment on the possibility of edge initiated epitaxy[34,35] or intercalation under the graphene[36,37], as has been observed in other systems.

After 2 ML of GaSb growth, nearly all of the holes have nucleated GaSb islands, which we further quantify in Supplemental Fig. 7. After 5 ML growth, the islands begin to grow laterally (Fig. 4a). At 50–100 ML, the islands extend and coalesce (Fig. 4b). Finally, after 200 ML a smooth GaSb film surface is recovered across length scales of several microns, with an atomic step-and-terrace morphology observed by AFM. Our results indicate that the holes created by native oxide desorption serve as sites for the nucleation, lateral epitaxy, and coalescence of GaSb films (Fig. 4c).

As a control, we also perform GaSb film grown on a graphene/ GaSb (001) sample for which we do not remove the interfacial oxides. For this sample, we anneal to 350 °C, which is below the native oxide desorption temperature. We find that the resulting GaSb film grown at 350 °C is polycrystalline (Supplementary Fig. 9). Therefore we conclude that removal of the native oxides is a crucial step for growing an epitaxial film.

Despite the presence of holes, GaSb films grown by a seeded lateral epitaxy mode can be exfoliated, as shown in the large scale SEM images of Fig. 1e, f. To understand this exfoliation further, in Fig. 5 we show higher magnification SEM images of the substrate after GaSb membrane exfoliation. In Fig. 5(a) we observe elongated clusters of spalling marks with width ~300 nm

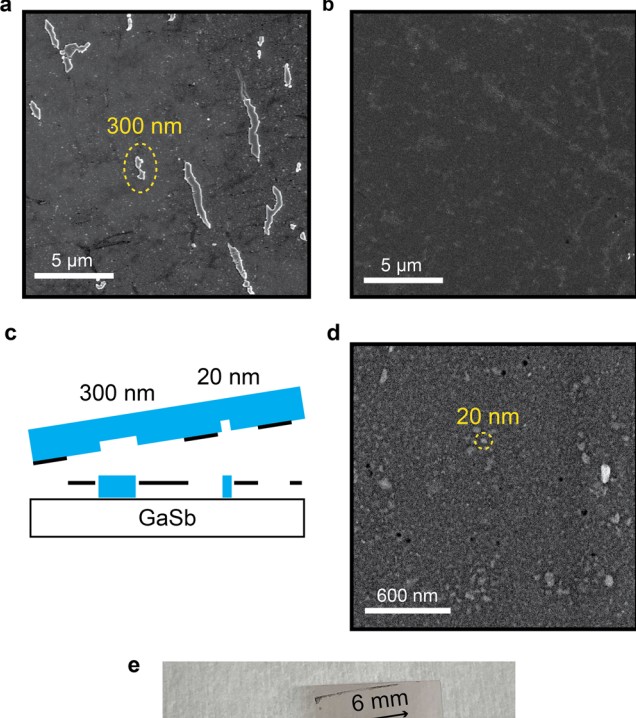

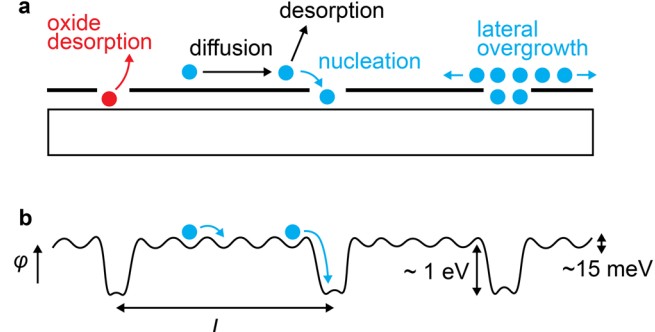

**Fig. 6 Fast surface diffusion favors seeded lateral epitaxy at pinholes.** **a** Pinholes are created by the native oxide desorption, which then serve as nucleation sites for epitaxial growth. **b** Schematic potential $\phi$ for adatoms diffusing on the graphene-terminated surface, with diffusion length $\lambda$. Seeded lateral epitaxy is expected when $L << \lambda$, where $L$ is the spacing between pinholes. Remote epitaxy requires $L > \lambda$.

**Fig. 5 Exfoliation of GaSb membranes that are seeded at pinholes. a** SEM image of the substrate after exfoliation, showing ~300 nm diameter spalling marks and clusters of spalling marks. **b** SEM image of the substrate showing a relatively smooth region with no large spalling marks. **c** Cartoon of the exfoliation. **d** Magnified SEM image of the substrate showing ~20 nm spalling marks. **e** Photograph of the substrate and the exfoliated GaSb membrane supported on thermal release tape. The lateral dimensions of the membrane are 5 mm by 6 mm.

and length ranging from 300 nm to several microns, which have approximately the same size and areal density as the 300 nm diameter holes observed in Fig. 3(a). We attribute these spalling marks to direct epitaxy in the 300 nm holes, where the film remains stuck to the GaSb substrate rather than being peeled away with the film/membrane. Other regions of the substrate appear relatively smooth at this length scale (Fig. 5(b)), since the original 300 nm holes from oxide desorption are sparsely distributed across the sample. Zooming into this relatively smooth region, we observe smaller scale ~20 nm islands (Fig. 5(d)), which we attribute to the film sticking at the ~20 nm graphene pinholes. These pinholes are typically evenly distributed across the sample after oxide desorption. After exfoliation we detect Raman 2D and G modes on both the exfoliated membrane side and on the substrate side, indicating that exfoliation tears the graphene (Supplementary Fig. 10).

**Comparison with remote epitaxy.** Our experiments reveal that pinhole defects drive the seeded lateral epitaxy of GaSb on graphene-terminated GaSb (001). These pinholes are created by native oxide desorption from the substrate. The resulting GaSb films have similar characteristics and structural quality as previous reports of remote epitaxy[1,3,5,6], namely, epitaxial alignment

to the underlying substrate, smooth atomically stepped surfaces, and the ability to exfoliate macroscopically smooth films. We understand the growth in terms of a balance between several atomic scale processes, as illustrated in Fig. 6. This balance dictates whether the growth is in a seeded lateral epitaxy regime or a remote epitaxy regime.

Seeded lateral epitaxy occurs in the graphene/GaSb system because (1) graphene is chemically inert and (2) the surface diffusion length $\lambda$ is long compared to the average spacing $L$ between pinhole defects ($\lambda >> L$). Previous experiments on confinement heteroepitaxy demonstrate that the effective surface diffusion length for group-III metal adatoms on graphene is of order 10 microns, since metal adatoms preferentially nucleate at graphene defects over length scales of order 10 microns[36,37]. Our preliminary experiments on lithographically defined graphene stripes reveal a similar picture (Fig. 7). We find that during the early stages of nucleation for GaAs films on patterned graphene/ Ge(001), the GaAs preferentially nucleates at exposed regions of the Ge substrate, even when the spacing between graphene openings is 10 microns. This $\lambda \sim 10$ micron effective diffusion length is much larger than the $L \sim 70$ nm spacing between pinhole defects on graphene/GaSb(001). Therefore, adatoms can sample enough of the surface to nucleate at pinhole defects, rather than on the inert graphene. Remote epitaxy would require a larger separation between defects, $L > \lambda$.

The long effective surface diffusion length arises from a combination of long surface diffusion and a high adatom desorption rate (Fig. 6a), since the sticking coefficient of Ga and Sb adatoms on inert masks at 500 °C is less than unity. Similar behavior has been observed for selective area epitaxy of III-V films on patterned dielectric masks, such as $SiO_2$ and $SiN_x$[14].

The relative size of potential modulations also explains the preferred nucleation and growth from pinholes (Fig. 6b). On graphene-terminated regions, the combined potential modulations from the graphene itself and from the substrate that permeate through graphene are expected to be weak. Density functional theory calculations suggest these modulations are of order $\Delta\phi \sim 15$ meV[6], which is smaller than the thermal energy $k_B T \sim 70$ meV at growth temperature 500 °C. An estimate of the free carrier screening from graphene also suggests this potential should be weak. For a free electron gas with density $10^{19} - 10^{20}$ cm$^{-3}$, the same density as graphene (graphite)[38], the Thomas-Fermi screening length is approximately $1/k_0 \sim 2$ Å. This is less than the film/substrate layer spacing of ~5 Å[1], suggesting that the screened potential above the graphene is exponentially weak. In

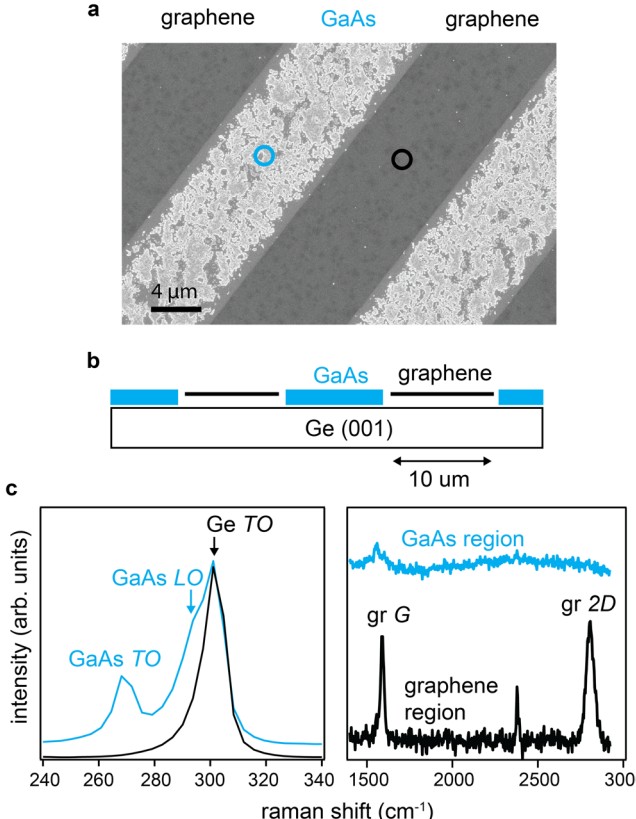

**Fig. 7 GaAs selectivity on patterned graphene/Ge.** The graphene was grown directly on the Ge (001) by CVD, thus avoiding the damage and interfacial oxides typically present for graphene layer transfers. We use photolithography and an oxygen plasma etch to define 10 micron wide stripes of alternating graphene and exposed Ge substrate. GaAs films were then grown on the patterned graphene substrate by MBE. **a** SEM image of GaAs nucleation on patterned graphene on a Ge (001) substrate. We observe selective nucleation of GaAs (light color) on the exposed Ge regions of the substrate, and no GaAs nucleation on the graphene. **b** Cartoon of the sample. **c** Raman spectra on the graphene masked region (black curves) and on the GaAs nucleation region (blue). Representative areas for these spectra are marked by the circles in the SEM image (**a**).

contrast, the potential modulation at a graphene pinhole is expected to be much larger, $\Delta\phi \sim 1$ eV, since this involves adatoms making chemical bonds to the graphene or to the underlying substrate. The pinholes provide the low energy sites for nucleation and growth. Epitaxial registry is obtained because the pinholes are locations of direct epitaxy to the substrate. Given these factors that favor seeded lateral epitaxy when there is a finite density of graphene defects ($L \ll \lambda$), our results suggest that previous demonstrations of remote epitaxy can alternatively be explained by a seeded lateral epitaxy mechanism.

Our experiments demonstrate that in the presence of a high concentration of graphene defects ($L \ll \lambda$), the defects dominate the growth mechanism. Due to the interfacial oxides, we expect that most graphene that is transferred to a foreign substrate will lie in this high defect density regime. Characterization of the graphene quality before the pre-growth anneal (Fig. 3 (350 °C) and Supplementary Fig. 8a) is not a reliable representation of the surface quality immediately prior to III-V film growth (Fig. 3 (450 °C) and Supplementary Fig. 8b). Additionally, the ability to exfoliate membranes from graphene is not sufficient to prove a remote epitaxy mechanism, since films grown by seeded lateral epitaxy can also be exfoliated (Fig. 5).

On the other hand, our results do not globally rule out the possibility of a "remote epitaxy" mechanism. The key requirement for testing a remote mechanism is to start with clean graphene that has low defect density ($L > \lambda$) immediately prior to film growth, i.e. including the pre-growth annealing. Graphene that is directly grown on the substrate of interest, rather than transferred, provides a cleaner starting point for such studies. III-nitride[36,39] and metal films[37] on epitaxial graphene/SiC, and III-arsenide films on CVD graphene/Ge (Fig. 7) are promising options. Our microscopic measurements set a higher experimental bar for proof of remote epitaxy.

## Discussion

Our in-situ experiments provide a fundamental understanding of the atomic scale growth mechanisms on graphene, which are crucial for controlling the structure, properties, and scalability of the resultant films and membranes. In addition, seeded lateral epitaxy on graphene offers several advantages over both conventional ELO and remote epitaxy.

Compared to conventional ELO, which uses thick dielectric masks, graphene is the atomically thin limit for a mask material. This thinness is attractive for applications where an electronically transparent interface is desired, e.g. a tunnel barrier[19,20]. Graphene masks also enable the etch-free synthesis and exfoliation of crystalline membranes, for applications in flexible electronics and for inducing novel properties via strain[5].

Compared to remote epitaxy, which is expected to only produce epitaxial films on substrates with polar bonding[6], seeded lateral epitaxy can produce single-crystalline films on nonpolar substrates[15]. Seeded lateral epitaxy is also tolerant to an imperfect graphene/substrate interface. Despite the oxides present at the graphene/substrate interface after graphene transfer, epitaxial growth can still be achieved by desorbing the native oxide to create pinholes. This tolerance is crucial for translating epitaxy on graphene-terminated surfaces to wafer scale, since it is difficult to perform native oxide-free graphene layer transfers at wafer scale.

Our work suggests that the controlled patterning of nanoscale openings in graphene (Fig. 7) provides a route to precisely engineer the size, location, and spatial distribution of nucleation sites, and resulting properties of coalesced films and membranes. Recently, patterned openings in a graphene mask have been used for the selective area epitaxy of 2D hexagonal-BN[21], and point defects in h-BN have have been used to seed 2D MoS₂ films with controlled orientations[22]. Our results show that openings in a graphene mask are also promising for the growth of three-dimensional materials like III-V semiconductors, by enabling their exfoliation into free-standing quasi-2D membranes.

## Methods

**Graphene synthesis and transfer**. Our graphene synthesis and transfer procedure is a modified polymer-assisted wet transfer of CVD-grown graphene[40], similar to the transfer recipe from previous demonstrations of remote epitaxy[1,6]. Graphene was grown by thermal chemical vapor deposition (CVD) of ultra high purity CH₄ at 1050 °C on Cu foil, as described in refs. [5,41]. The graphene/Cu foils were then cut and and flattened using clean glass slides to match the dimensions of the semiconductor substrate. Approximately 200 nm of 950 K C2 PMMA (Chlorobenzene base solvent, 2% by wt., Kayaku Advanced Materials, Inc.) was spin coated on the graphene/Cu foil at 2000 RPM for 2 minutes and left to cure at room temperature for 24 hours. Graphene on the backside of the Cu foil was removed via reactive ion etching using 90 W O₂ plasma at a pressure of 100 mTorr for 30 s. The Cu foil was then etched by placing the PMMA/graphene/Cu foil stack on the surface of an etch solution containing 1-part ammonium persulfate (APS-100, Transene) and 3-parts H₂O. After 16 hours of etching at room temperature, the floating PMMA/graphene membrane was scooped up with a clean glass slide and sequentially transferred onto five 5-min water baths to rinse any etch residuals.

GaSb (001) substrates were etched in 38% HCl solution for 5 min followed by a 2-propanol rinse to remove some of the native oxides. We then used the etched GaSb substrate to scoop the PMMA/graphene stack from the final water bath. To remove water at the graphene/substrate interface, the samples were baked in air at 50 °C for

15 minutes, then at 150 °C for another 15 min. The PMMA was dissolved by submerging the sample in an acetone bath at 80 °C for 3 h. This is followed by an IPA rinse and nitrogen drying. The sample is indium bonded onto a molybdenum puck and outgassed at 150 °C for 2 h in a loadlock at a pressure $p < 5 \times 10^{-7}$ Torr before introduction to the MBE growth chamber. Finally, the sample is annealed at ~350 °C for 1 h in the MBE chamber to desorb organic residuals.

**MBE growth of GaSb**. GaSb films were grown in a custom molecular beam epitaxy chamber (Mantis Deposition) using an effusion cell for Ga and a thermal cracker cell for Sb (MBE Komponenten). Temperatures were measured using a pyrometer that is calibrated to the native oxide desorption temperature of GaAs. In-situ reflection high energy electron diffraction (RHEED) measurements were performed using a Staib RHEED gun at 15 keV.

**In-situ XPS**. X-ray photoemission spectroscopy (XPS) measurements were performed in an XPS chamber that is connected to the MBE via an ultrahigh vacuum ($p < 5 \times 10^{-10}$ Torr) transfer chamber, such that samples are transferred in vacuum and are not exposed to air. We use a non-monochromated Al $K_{\alpha}$ x-ray source (1486.6 eV) and an Omicron EA125 hemispherical analyzer with an energy resolution of 1.08 eV. Samples were annealed in the MBE and measured in the XPS at room temperature.

**AFM, SEM, and Raman spectroscopy**. Field emission scanning electron microscopy (SEM) was performed using a Zeiss GeminiSEM 450. Raman spectroscopy was performed using a 532 nm wavelength laser (Thermo Scientific DXR Raman Microscope). The laser power is kept below 5 mW in order to prevent damage to the graphene. In order to acquire representative raman spectra, 2600 spectra where captured and averaged over a 2500 $\mu m^2$ area. Atomic force microscopy (AFM) measurements were performed using a Bruker Dimension Icon in tapping mode.

**X-ray diffraction**. X-ray diffraction measurements were performed using a four-circle Malvern Panalytical Empyrean diffractometer, using Cu $K\alpha$ radiation. Symmetric $\omega - 2\theta$ measurements ($\omega$ is the sample tilt in the scattering plane and $2\theta$ is the Bragg angle) and azimuthal pole figure ($\phi$) scans were performed using a 4 bounce Ge 220 monochromator on the incident radiation. Rocking curve measurements were performed using an additional 3 bounce Ge 220 analyzer crystal.

**GaSb exfoliation**. To exfoliate the GaSb epilayer, a 100 nm Ni stressor layer was deposited on the GaSb / graphene / GaSb (001) heterostructure at room temperature using a metal evaporator (Angstrom Engineering Inc.) in high-vacuum ($p < 2 \times 10^{-6}$ Torr). The film was then exfoliated with two methods. Crystalbond 509 adhesive, with a flow point of 120 °C, was gently smeared on glass slide on a hot plate at 150 °C. The sample is then placed film-side down on the adhesive and allowed to cool. The membrane can be easily exfoliated by gently dropping the glass slide from a height of approximately one foot onto a clean surface. The force from the drop is enough to release the substrate from the membrane, leaving the membrane adhered on the glass slide. Thermal release tape (Revalpha 3195H, Semiconductor Equipment Corp.) was also used to exfoliate by stamping onto the sample, and then carefully peeling the tape away from the substrate.

**Photoluminescence measurements**. Steady-state photoluminescence (PL) measurements were performed at 300 K with a CW Ar-ion pump laser at a wavelength of 514.5 nm, using a 1500 nm (600 gr/mm) grating and a liquid-nitrogen cooled Ge detector. The power density was approximately 6.96 watts/cm$^2$.

## Data availability

All data generated in this study have been deposited in the Zenodo database under accession code https://doi.org/10.5281/zenodo.6596372.

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

## Acknowledgements
We thank Thomas F. Kuech for helpful discussions and for donating his XPS system. We thank Seth R. Bank and Hari P. Nair for discussions on photoluminescence measurements. GaSb film growth and in-situ XPS were supported primarily by the National Science Foundation, award number DMR-1752797 (JKK). Graphene transfers and III-V growth on patterned graphene were supported by the Defense Advanced Research Projects Administration, DARPA Grant number D19AP00088 (JKK). Graphene synthesis and characterization are supported by the U.S. Department of Energy, Office of Science, Basic Energy Sciences, under award no. DE-SC0016007 (MSA). PL measurements were supported by the National Science Foundation ECCS-1806285 (LJM). We gratefully acknowledge the use of Raman, electron microscopy, and PPMS facilities supported by the NSF through the University of Wisconsin Materials Research Science and Engineering Center under Grant No. DMR-1720415. Use of the PPMS was also supported by the Wisconsin Alumni Research Foundation WARF (JKK).

## Author contributions
J.K.K. conceived the project. S.M. performed graphene layer transfer, GaSb film growth and exfoliation, and characterization of structure and morphology. P.S. performed in-situ XPS measurements. Z.L. aided with structural characterization and performed GaAs growth on graphene/Ge. V.S. and M.S.A. synthesized the graphene. D.D. aided with graphene and GaSb film characterization. S.X. and N.P. performed photoluminescence spectroscopy under the supervision of L.M. J.K.K. and S.M. wrote the manuscript with input from M.S.A. and all the authors.

## Competing interests
The authors declare no competing interests.
