## [Peer Review File · Nature Communications]

Pinhole-seeded lateral epitaxy and exfoliation of GaSb films on graphene-terminated surfacesREVIEWER COMMENTS

Reviewer #1 (Remarks to the Author):

This paper reports on the homoepitaxial lateral overgrowth of a GaSb film using a graphene mask layer. For seeding, pin holes were formed on graphene during annealing process of a GaSb substrate with a native oxide layer. Since graphene has previously been used as a pattern mask layer for the lateral overgrowth of other semiconductor films, I think that the originality of this manuscript is based on the use of pinholes naturally formed by thermal annealing of the oxide layer as a pattern mask. Since the pinholes can be formed by simple thermal annealing without using any conventional lithography technique, it may reduce cost to produce the films. However, the crystal quality of the film is not investigated. According to the SEM image, it seems that there are so many defects in the film that the authors should report on the crystal quality, electrical and optical characteristics of the films.

Further comments:

1. What is the maximum size of the free standing film?
2. How can you control the pinhole size and density to get high quality films?
3. How can you prove that the films are not grown by remote epitaxy?

Reviewer #2 (Remarks to the Author):

The manuscript entitled "Pinhole-seeded lateral epitaxy and exfoliation on graphene-terminated surfaces" reports on (remote) epitaxy studies of GaSb on graphene-terminated GaSb(001) templates. Remote epitaxy is a topic of current interest given the great potential of this method for large-scale synthesis of 2D and 3D materials and their heterostructures on arbitrary substrates, including flexible ones. The results presented in this work are highly relevant as the authors demonstrate that GaSb epitaxy proceeds in fact via defect-assisted nucleation and lateral growth, and not due to a remote interaction with the GaSb substrate below graphene. This is significant as it advances the state of the art regarding our current understanding of the so-called remote epitaxy process. Hence, this has great implications for wafer-scale synthesis and transfer of many other materials, in particular because the main ingredient for successful epitaxy seems to be defect engineering in graphene, a material which can in principle be transferred to virtually any crystalline substrate. The methodology utilized by the authors is completely adequate with detailed information being provided.

Nevertheless, it is my opinion that before the manuscript receives further consideration, it should be revised/improved concerning two aspects. First, the authors should add additional experimental data and analysis to give further support to their conclusions and claims (e.g., as supplementary material). Second, the authors should cite appropriate literature which is now missing. Please see below a list of specific issues that must be addressed by the authors.

1-Given the fact that defect formation in graphene plays a central role in the process, the authors should provide more detailed investigations regarding this aspect, e.g. via Raman spectroscopy. For instance, for sample annealed at 450 C, although the increase of the D peak appears to be minimum, other features of the Raman spectra should be analyzed, such as the FWHM and position of the 2D and G peaks, 2D/G intensity (area) ratio. They can be used to estimate the defect density as a function of annealing temperature and eventually strain. The latter might evolve as well during oxide desorption e.g. due a change in surface corrugation.

2-From the AFM images (Fig.3), it seems that the holes which start to form in graphene at 450C are usually located close to surface protrusions from the "bumpy" GaSb surface. The authors should discuss this in more detail. Are these protrusions the remaining oxide "islands" that desorbs then during annealing and "open a hole" in the uppermost graphene? What is the possible

chemical process allowing for the graphene “etching” in these areas? As the annealing temperature increases, is there a change (decrease) in the surface roughness e.g. due to a completed Sb-oxide desorption?

3-Some of the questions above can eventually be answered by a more detailed analysis of the XPS results. Although the authors utilized a non-monochromated X-ray source, fitting of the C1s/O1s spectral region could shine some light on the chemical configuration.

4-It is also important to provide more data/discussion on how nucleation takes place, to include the potential scenarios. Does it proceed solely as illustrated in Fig. 5c, i.e. GaSb nuclei grow on top of the GaSb substrate? Could it be that lateral seeding at graphene edges takes place as well? This has an enormous implication on further development of defect engineering in graphene (e.g. what is the minimum defect size needed) and on the quality of the transfer process, as well as on the reusability of the host graphene/substrate.

5-The authors should show more results for growth experiments on “control” samples, for instance for the sample annealed at 350C in which graphene is still “defect-free”. What happens in this case? This is a pertinent question every reader would probably like to ask.

6-GaSb (homoepitaxial) growth below graphene via Ga/Sb diffusion through defects could in principle also take place (as shown in Ref. 31 for GaN). Have the authors considered this aspect? In other words, are there results that support/rule out this possibility?

7-In Fig. 5c it is shown that the graphene remains at the substrate after transfer of the GaSb membrane. Please include additional results (e.g. Raman) that support this claim. This is not clear by only looking into the SEM images of Fig. 5. In that respect, please show data or at least comment on the prospect of reutilizing the host substrate for the consecutive synthesis of more GaSb membranes.

8-The schematic illustration in Fig. 3d is misleading as it gives the impression that nanoholes already present in graphene allow for the out-diffusion of species from the substrate. Although this might also occur, this is not what is intended to be shown. Thus, I suggest that the authors improve this illustration.

9-Last but not least: by reading the abstract, introduction, and discussion, one may get the idea that the concept of using defects in graphene as a “powerful knob for controlling epitaxial growth on 2D materials surface” is something totally new and suggested for the first time. This is not the case and thus I recommend the authors to include literature showing state-of-the-art results in this topic. See for instance <https://www.nature.com/articles/s41699-021-00250-z> as well as <https://journals.aps.org/prb/abstract/10.1103/PhysRevB.99.155430>

Reviewer #3 (Remarks to the Author):

This is an exciting and thorough paper that clarifies the underlying growth physics that underpins remote epitaxy. I think it will attract significant interest from a wide range of audiences and is a good fit with Nature Communications. I believe it will generate much discussion, given the nature of the findings, but I believe the conclusions are well supported – it will drive much subsequent research and I believe the findings will be validated by this follow-on work. This manuscript is essentially two papers in one as it:

1. Demonstrates the presence of pinholes in the graphene and tracks their formation back to the thermal treatments prior to growth (e.g. Fig. 3), as well as shows nucleation at those point defects (e.g. Fig. 4). The latter point is really driven home by Fig. S-6, which is essentially the third most important figure in the paper.
2. Clarifies that the density of pinholes is sufficiently high and the driving forces to preferential nucleation are sufficiently strong (or put another way, the diffusion lengths are sufficiently high)

that pin-hole seeded nucleation can fully explain the remote epitaxy process. This is the more controversial result; however, I think that the experiments reported in Fig. S-6 really strengthen those in Fig. 4, as well as the diffusion length arguments in Section II.D, so I believe the authors' conclusions will ultimately be proven correct. Fig. S-6 directly connects remote epitaxy to the established LEO MBE growth mode identified by Nishinaga and provides compelling evidence of the authors' interpretation of the findings.

A few minor things:

1. As mentioned above, I think Fig. S-6 is very important to the paper; while I see why it's not included in the main manuscript, I think having it in the main body could bring significant benefits (though I'm not sure if that's feasible given things like length constraints).
2. The authors do not discuss XRD in Methods. This adds a little bit of ambiguity; for example, is Fig. 1d an omega-2theta or a rocking curve (the plot is missing an axis label)?
3. I would suggest plotting Fig. 1d on a log scale as that is a better way to convey that there are not things like alternate phases buried at low intensities. I might also suggest adding the XRD of the corresponding GaSb-on-GaSb homoepitaxy from the other half of the sample. This will help the reader compare the relative quality of the films (since there's no electrical or optical data like photoluminescence). Ideally a pole figure would be better than a wide angle scan, but the wide angle scan is fine.
4. Along those same lines, I wonder if rocking curves might be better for Fig. 1g (comparison of the FWHM will then give you useful insights into relative crystal quality – the omega-2thetas shown in the manuscript should serve that purpose, but only if the films are exactly the same thickness, which is maybe not a given).
5. In the Fig. 1 caption, part (d) is mislabeled as (g).

Some suggestions for follow-up experiments (not for this paper, but just some possibly useful ideas):

1. If you vary the pinhole size/density through pre-growth thermal/chemical treatments, you should be able to see differences in the fracture strength / interfacial adhesion energy. The ideal person for these kinds of measurements would be someone like Ken Liechti.
2. With direct attachment to the substrate comes the possibility of critical thickness in heteroepitaxy, which you might see at high pinhole densities and strains. There's also the possibility (though this part is speculative on my part) of hitting the compliant substrate criteria for infinite critical thickness at sufficiently low levels of pinholes/mismatch. That would be enabling to say the least!
3. I don't think it is necessary for this paper, but a cross-sectional TEM through a sufficiently large pinhole would be very helpful.

REVIEWER COMMENTS

Reviewer #1 (Remarks to the Author):

This paper reports on the homoepitaxial lateral overgrowth of a GaSb film using a graphene mask layer. For seeding, pin holes were formed on graphene during annealing process of a GaSb substrate with a native oxide layer. Since graphene has previously been used as a pattern mask layer for the lateral overgrowth of other semiconductor films, I think that the originality of this manuscript is based on the use of pinholes naturally formed by thermal annealing of the oxide layer as a pattern mask. Since the pinholes can be formed by simple thermal annealing without using any conventional lithography technique, it may reduce cost to produce the films. However, the crystal quality of the film is not investigated. According to the SEM image, it seems that there are so many defects in the film that the authors should report on the crystal quality, electrical and optical characteristics of the films.

We thank the reviewer. Regarding comparisons of film quality, on page 3 of the manuscript we add: "In the rocking curves of the 004 reflection for a 120 nm GaSb film on graphene/GaSb (Supplemental Fig. S-1), in addition to the expected peak at $w = 0$, we observe a second peak shifted by $w = -15$ arc seconds. The full width at half maxima (fwhm) of these two peaks are 10.44 and 12.59 arc seconds, respectively, compared to a width of 6.74 arc seconds for a homoepitaxial film." We also add pole figure phi scans of the 202 reflection, which show the expected 4-fold symmetry of an epitaxial film that is oriented in-plane (Supplemental Fig. S-1).

It is important to note that previous reports of III-V remote epitaxy do not show rocking curves. Additionally, the previous reports are for much thicker films (5-10 microns), compared to our ~100 nm films that focus on the initial stages of nucleation. Therefore, we are not able to make a direct comparison of the structural quality of our GaSb films with the structural quality of other "remote epitaxy" reports from the literature, including GaAs [Y. Kim, et. al. Nature 544, 340-343 (2017)], InGaP [S-H Bae, et. al. Nature Nano. 15, 272-276 (2020)], GaN [W. Kong, et. al. Nature Materials 17, 999-1004 (2018)], or halide perovskites [J. Jiang, et. al. Nature Commun. 10, 4145 (2019)]. The only published rocking curve we are aware of is for GdPtSb films grown on graphene/Al₂O₃ [D. Du, et. al. Nature Commun. 12, 2494 (2021)]. For GdPtSb fwhm is 15 arc seconds, compared to 6 arc seconds for the Al₂O₃ substrate.

We hope that these results will motivate further quantitative analysis of the structure and defects in films grown on graphene. A recent TEM study showed that InGaP films grown on graphene/GaAs (001) show lattice rotations with respect to the GaAs substrate, whereas InGaP grown directly in GaAs does not [S-H Bae, et. al. Nature Nano, 15, 272-276 (2020)],

Supplemental Fig 5]. The presence of rotational domains may explain both the wider rocking curves and the peak splitting for films grown on graphene.

Further comments:

1. What is the maximum size of the free standing film?

The examples shown in Figures 1 and 5 show free-standing membranes with lateral dimensions of 10 x 6 mm and 5 x 6 mm, respectively. We expect that the maximum size of exfoliatable films will ultimately be limited by the size of graphene that can be transferred onto the epi-ready substrate. Several groups have reported graphene transfer at wafer scale (3" to 8") with minimal fold and wrinkles [L. Gao et. al., Nature 505, 190-194 (2014); W-H Lin, et. al. ACS Nano 8, 2, 1784-1791 (2014); Y. Lee, et. al. Nano Lett 10, 2, 490-493 (2010)]. Therefore we expect that exfoliation at a 3-8 inch scale may be possible. However, since the focus of the current manuscript was to understand the atomic-scale growth mechanisms on graphene, we have not actively pursued scale-up.

2. How can you control the pinhole size and density to get high quality films?

This is an excellent direction. We are actively investigating control of pinhole size and density now. The key is to control the graphene/substrate interface cleanliness. Transferred graphene is difficult due to the interfacial oxides and other contaminants. We find that graphene grown directly on the substrate of interest provides the best starting point for intentionally patterning pinholes via optical or e-beam lithography. Graphene grown directly on Ge substrates is a good platform for these studies, since graphene can be grown directly on Ge by CVD, and Ge is lattice matched to GaAs films of interest.

Changes to manuscript: on page 8 we add: "Graphene that is directly grown on the substrate of interest, rather than transferred, provides a cleaner starting point for such studies."

3. How can you prove that the films are not grown by remote epitaxy?

Our experiments demonstrate that graphene pinholes, which are created by the native oxide desorption, seed the lateral epitaxy of GaSb films on graphene/GaSb (001). Seeded lateral epitaxy occurs because of the strong driving force for nucleation at the exposed pinholes, since (1) graphene is chemically inert and (2) surface diffusion on graphene is quite fast. Remote interactions are not needed in order to explain the resulting epitaxial films.

We caution that our results do not strictly mean that a "remote epitaxy" mechanism is impossible. Rather, our data show that due to the high concentration of pinholes, remote epitaxy is not the dominant mechanism in most experimental implementations to date. We follow the same graphene layer transfer as previous reports and provide the first microscopic characterization of the graphene/substrate surface in its state immediately prior to growth, i.e. after annealing up to the III-V growth temperature. This annealing procedure creates the pinholes, which notably are not present before the anneal. As discussed in Section II.D "comparison with remote epitaxy" we argue that a remote epitaxy mechanism still may be

possible on graphene samples with much lower defect density, i.e. $L > \lambda$ where L is the spacing between pinholes and λ is the surface diffusion length. We think that growth on clean graphene with $L > \lambda$ will be a fruitful line of research.

Changes to the manuscript: we clarify the possibility of remote epitaxy at the bottom of page 7: “We caution that our results do not strictly rule out the possibility of remote epitaxy...”

Reviewer #2 (Remarks to the Author):

Dear Editor,

The manuscript entitled “Pinhole-seeded lateral epitaxy and exfoliation on graphene-terminated surfaces” reports on (remote) epitaxy studies of GaSb on graphene-terminated GaSb(001) templates. Remote epitaxy is a topic of current interest given the great potential of this method for large-scale synthesis of 2D and 3D materials and their heterostructures on arbitrary substrates, including flexible ones. The results presented in this work are highly relevant as the authors demonstrate that GaSb epitaxy proceeds in fact via defect-assisted nucleation and lateral growth, and not due to a remote interaction with the GaSb substrate below graphene. This is significant as it advances the state of the art regarding our current understanding of the so-called remote epitaxy process. Hence, this has great implications for wafer-scale synthesis and transfer of many other materials, in particular because the main ingredient for successful epitaxy seems to be defect engineering in graphene, a material which can in principle be transferred to virtually any crystalline substrate. The methodology utilized by the authors is completely adequate with detailed information being provided.

We thank Reviewer 2 for the positive remarks on impact and methodology.

Nevertheless, it is my opinion that before the manuscript receives further consideration, it should be revised/improved concerning two aspects. First, the authors should add additional experimental data and analysis to give further support to their conclusions and claims (e.g., as supplementary material). Second, the authors should cite appropriate literature which is now missing. Please see below a list of specific issues that must be addressed by the authors.

Thank you for this suggestion. As detailed below, we have added additional experimental data and analysis in the supplemental materials. Thank you for also bringing the additional literature on 2D material masks to our attention. These additions strengthen the conclusions and broader context of this work.

1-Given the fact that defect formation in graphene plays a central role in the process, the authors should provide more detailed investigations regarding this aspect, e.g. via Raman spectroscopy. For instance, for sample annealed at 450 C, although the increase of the D peak appears to be minimum, other features of the Raman spectra should be analyzed, such as the FWHM and position of the 2D and G peaks, 2D/G intensity (area) ratio. They can be used to

estimate the defect density as a function of annealing temperature and eventually strain. The latter might evolve as well during oxide desorption e.g. due a change in surface corrugation.

We thank the reviewer for this suggestion. In Supplemental Fig. S-6 we provide a more detailed analysis of the Raman spectra as a function of anneal temperature. We summarize our analysis as follows:

- 2D/G intensity (area) ratio - This quantity decreases with anneal temperature, indicating an increasing concentration of defects
- 2D frequency vs G frequency - annealing produces changes in both strain and doping of the graphene. We estimate the amount of strain and doping from the shifts in the 2D and G frequencies, based on Ref. 42.

2-From the AFM images (Fig.3), it seems that the holes which start to form in graphene at 450C are usually located close to surface protrusions from the “bumpy” GaSb surface. The authors should discuss this in more detail. Are these protusions the remaining oxide “islands” that desorbs then during annealing and “open a hole” in the uppermost graphene? What is the possible chemical process allowing for the graphene “etching” in these areas? As the annealing temperature increases, is there a change (decrease) in the surface roughness e.g. due to a completed Sb-oxide desorption?

Thank you for this suggestion. We added a discussion of possible graphene etching mechanisms on page 5:

“The pinholes appear to be localized at the bumps that were observed after annealing to 350C. However, a more detailed study of samples before and after de-oxidation, at exactly the same location, would provide a more definitive answer to their origin. Oxygen (and oxygen plasmas) are well known to etch graphene [29], particularly at edges and extended defects like grain boundaries. In the present case, we expect that the larger strains around these protrusions may enhance the oxygen etch rate at these locations, compared to smoother regions of the graphene. Similar enhanced oxygen etching of graphene on other substrates has been attributed to roughness and impurities in the underlying substrate [30].”

Changes in surface roughness as a function of oxide desorption is an interesting idea. As a first estimate, we find that the total room mean squared surface roughness from AFM decreases as a function of annealing:

As transferred - 1.169 nm
450 C - 0.7856 nm
550 C - 0.7396 nm

However, we caution that this surface roughness analysis is performed on the entire field of view of the AFM images in Fig. 3b, which includes the pinholes themselves. Therefore, the roughness of the graphene between the pinholes is smaller. We also caution that roughness metrics are sensitive to the length scale of normalization. Given these challenges, we believe that a quantitative analysis of the roughness is more appropriate for a later publication.

3-Some of the questions above can eventually be answered by a more detailed analysis of the XPS results. Although the authors utilized a non-monochromated X-ray source, fitting of the C1s/O1s spectral region could shine some light on the chemical configuration.

We thank the reviewer for this suggestion. We agree that a quantitative analysis of the XPS lineshapes may provide valuable information on the oxidation state of carbon and other species. Unfortunately, due to the limited resolution using our non-monochromated x-ray gun, we were not able to sufficiently constrain our fits for C 1s lineshape on graphene/GaSb (001). We think that more quantitative analysis using monochromated x-rays from a synchrotron is an excellent line of future work.

4-It is also important to provide more data/discussion on how nucleation takes place, to include the potential scenarios. Does it proceed solely as illustrated in Fig. 5c, i.e. GaSb nuclei grow on top of the GaSb substrate? Could it be that lateral seeding at graphene edges takes place as well? This has an enormous implication on further development of defect engineering in graphene (e.g. what is the minimum defect size needed) and on the quality of the transfer process, as well as on the reusability of the host graphene/substrate.

At this point, in the SEM images of large holes (~300 nm diameter) we have only observed nucleation inside the pinholes, directly on the exposed substrate (see additional SEM images below). Lateral seeding at graphene edges would be a very interesting result, but we have not directly observed that in any of our SEM studies. A few additional SEM images are shown below. It is also not clear from the AFM measurements of the smaller pinholes.

Changes to manuscript: on page 5 we add: “At this point we primarily observe nucleation on top of exposed regions of the GaSb substrate, rather than at graphene edges or underneath the graphene. Further experiments are required in order to comment on the possibility of edge initiated epitaxy [34, 35] or intercalation under the graphene [36, 37], as has been observed in other systems.”

5-The authors should show more results for growth experiments on “control” samples, for instance for the sample annealed at 350C in which graphene is still “defect-free”. What happens in this case? This is a pertinent question every reader would probably like to ask.

This is an excellent suggestion. We have added Supplemental Fig. S-8 and a note at the bottom of page 5, where we show diffraction measurements for a sample annealed at 350C, below the native oxide desorption temperature. A GaSb film is then grown at 350 C. Both the RHEED and XRD omega-2theta scans show that the GaSb film grown under these conditions is polycrystalline.

The result has a strong resemblance to Kim et. al Nature 544, 340-343 (2017), who reported that annealing the graphene/GaAs (001) substrates in H₂ prior to film growth was necessary in order to obtain epitaxial GaAs growth. We suspect that annealing in H₂ removes the interfacial graphene/GaAs oxides, similar to our results which show how UHV annealing removes graphene/GaSb oxides.

6-GaSb (homoepitaxial) growth below graphene via Ga/Sb diffusion through defects could in principle also take place (as shown in Ref. 31 for GaN). Have the authors considered this aspect? In other words, are there results that support/rule out this possibility?

Growth by intercalation at defects is an interesting concept. At this point our data is not sufficient to support or rule out this possibility. We added a brief comment on this possibility on page 5: “Further experiments are required in order to comment on the possibility of edge initiated epitaxy [34, 35] or intercalation under the graphene [36, 37], as has been observed in other systems.”

7-In Fig. 5c it is shown that the graphene remains at the substrate after transfer of the GaSb membrane. Please include additional results (e.g. Raman) that support this claim. This is not clear by only looking into the SEM images of Fig. 5. In that respect, please show data or at least comment on the prospect of reutilizing the host substrate for the consecutive synthesis of more GaSb membranes.

In the new Supplemental Figure S-9 we include Raman measurements of the film side and the substrate side, after exfoliation. We observe graphene 2D and G modes on both sides, indicating that the graphene has torn and appears on both sides. Re-use of the substrate will require transfer of another layer of graphene. We have updated Fig 5c to reflect the fact that after exfoliation, graphene is present on both sides.

8-The schematic illustration in Fig. 3d is misleading as it gives the impression that nanoholes already present in graphene allow for the out-diffusion of species from the substrate. Although this might also occur, this is not what is intended to be shown. Thus, I suggest that the authors improve this illustration.

Thank you. We have updated Fig 3d to emphasize that the pinholes are created by the oxide desorption.

9-Last but not least: by reading the abstract, introduction, and discussion, one may get the idea that the concept of using defects in graphene as a “powerful knob for controlling epitaxial growth on 2D materials surface” is something totally new and suggested for the first time. This is not the case and thus I recommend the authors to include literature showing state-of-the-art results in this topic. See for instance <https://www.nature.com/articles/s41699-021-00250-z> as well as <https://journals.aps.org/prb/abstract/10.1103/PhysRevB.99.155430>

Thank you for bringing these references to our attention. We add the following discussion to provide a greater context to the use of patterned 2D masks:

Abstract (pg 1): “ Our results show that the controlled introduction of nanoscale openings in graphene provides a powerful route towards tuning the growth and properties of 3D epitaxial films and membranes on 2D material masks”

Introduction (pg 2): “Our experiments demonstrate that the introduction of defects provides a powerful knob for controlling epitaxial growth of covalent 3D materials on 2D materials surfaces, similar to recent work on growth of 2D materials on patterned 2D masks [21, 22].”

Conclusions (pg 8): “Recently, patterned openings in a graphene mask have been used for the selective area epitaxy of 2D hexagonal-BN [21], and point defects in h-BN have have been used to seed 2D MoS₂ films with controlled orientations [22]. Our results show that openings in a graphene mask are also promising for the growth of three-dimensional materials like III-V semiconductors, by enabling their exfoliation into free-standing quasi-2D membranes.”

Reviewer #3 (Remarks to the Author):

This is an exciting and thorough paper that clarifies the underlying growth physics that underpins remote epitaxy. I think it will attract significant interest from a wide range of audiences and is a good fit with Nature Communications. I believe it will generate much discussion, given the nature of the findings, but I believe the conclusions are well supported – it will drive much subsequent research and I believe the findings will be validated by this follow-on work. This manuscript is essentially two papers in one as it:

1. Demonstrates the presence of pinholes in the graphene and tracks their formation back to the thermal treatments prior to growth (e.g. Fig. 3), as well as shows nucleation at those point defects (e.g. Fig. 4). The latter point is really driven home by Fig. S-6, which is essentially the third most important figure in the paper.
2. Clarifies that the density of pinholes is sufficiently high and the driving forces to preferential nucleation are sufficiently strong (or put another way, the diffusion lengths are sufficiently high) that pin-hole seeded nucleation can fully explain the remote epitaxy process. This is the more controversial result; however, I think that the experiments reported in Fig. S-6 really strengthen those in Fig. 4, as well as the diffusion length arguments in Section II.D, so I believe the authors’

conclusions will ultimately be proven correct. Fig. S-6 directly connects remote epitaxy to the established LEO MBE growth mode identified by Nishinaga and provides compelling evidence of the authors' interpretation of the findings.

We thank Reviewer 3 for the encouraging feedback.

A few minor things:

1. As mentioned above, I think Fig. S-6 is very important to the paper; while I see why it's not included in the main manuscript, I think having it in the main body could bring significant benefits (though I'm not sure if that's feasible given things like length constraints).

We agree that Fig. S-6 is very important to this paper. We have decided to include it in the main manuscript as Fig. 7. We have checked that with this addition, the manuscript is still within the 5000 word, 10 display item limit for Nature Communications.

2. The authors do not discuss XRD in Methods. This adds a little bit of ambiguity; for example, is Fig. 1d an omega-2theta or a rocking curve (the plot is missing an axis label)?

Thank you for pointing out this ambiguity. We have added a description of XRD methods on page 9 and have added any missing labels.

3. I would suggest plotting Fig. 1d on a log scale as that is a better way to convey that there are not things like alternate phases buried at low intensities. I might also suggest adding the XRD of the corresponding GaSb-on-GaSb homoepitaxy from the other half of the sample. This will help the reader compare the relative quality of the films (since there's no electrical or optical data like photoluminescence). Ideally a pole figure would be better than a wide angle scan, but the wide angle scan is fine.

Thank you for pointing out this ambiguity. Fig 1d is indeed in log scale. We have added the missing axis labels for clarity.

4. Along those same lines, I wonder if rocking curves might be better for Fig. 1g (comparison of the FWHM will then give you useful insights into relative crystal quality – the omega-2theta shown in the manuscript should serve that purpose, but only if the films are exactly the same thickness, which is maybe not a given).

We agree that a rocking curve is an excellent metric for comparison between films grown on graphene and films grown by (direct) homoepitaxy. We add a comparison of rocking curves and a pole figure in Supplemental Fig. S-1. We also reference the rocking curve fwhm in the main text on page 3:

"In the rocking curves of the 004 reflection for a 120 nm GaSb film on graphene/GaSb (Supplemental Fig. S-1), in addition to the expected peak at $\omega = 0$, we observe a second

peak shifted by $\omega = -15$ arc seconds. The full width at half maxima (fwhm) of these two peaks are 10.44 and 12.59 arc seconds, respectively, compared to a width of 6.74 arc seconds for a homoepitaxial film.”

5. In the Fig. 1 caption, part (d) is mislabeled as (g).

Thank you. We have updated the caption.

Some suggestions for follow-up experiments (not for this paper, but just some possibly useful ideas):

1. If you vary the pinhole size/density through pre-growth thermal/chemical treatments, you should be able to see differences in the fracture strength / interfacial adhesion energy. The ideal person for these kinds of measurements would be someone like Ken Liechti.
2. With direct attachment to the substrate comes the possibility of critical thickness in heteroepitaxy, which you might see at high pinhole densities and strains. There's also the possibility (though this part is speculative on my part) of hitting the compliant substrate criteria for infinite critical thickness at sufficiently low levels of pinholes/mismatch. That would be enabling to say the least!
3. I don't think it is necessary for this paper, but a cross-sectional TEM through a sufficiently large pinhole would be very helpful.

Thank you for these suggestions!

REVIEWER COMMENTS

Reviewer #1 (Remarks to the Author):

The revised manuscript looks better although I suggest that the authors should give more clear answers to my questions. First of all, the electrical and optical characteristics of the free-standing film should be investigated for device applications. If the films do not have high quality, I would not like to recommend that this paper should be published in this journal. In addition, the growth mechanism is not still clear. If the remote epitaxy cannot be ruled out for the film growth as the authors mentioned, the authors' argument on the pin-hole seeded epitaxy and results could not be strong to be published in this journal.

Reviewer #2 (Remarks to the Author):

The authors implemented several changes in the revised version of the manuscript based on my (and the other reviewers) comments and criticism. Thus, it is my opinion that the manuscript has improved substantially and is now suitable for publication in Nature Communications.

Reviewer #3 (Remarks to the Author):

The authors have satisfactorily addressed my concerns and (I believe) those of the other reviewers as well. The paper is now much stronger than the initial submission and I believe it should be published in Nature Communications.

Reviewer #1 (Remarks to the Author):

The revised manuscript looks better although I suggest that the authors should give more clear answers to my questions. First of all, the electrical and optical characteristics of the free-standing film should be investigated for device applications. If the films do not have high quality, I would not like to recommend that this paper should be published in this journal.

We now include electrical transport and photoluminescence (PL) measurements to assess film quality (Supplementary Figures S-2 and S-3, mentioned in the main text page 3). In summary, both magnetotransport and PL measurements of films grown on graphene show similar carrier mobility and PL lineshapes as direct epitaxial films or substrates without graphene. We describe the key results below:

Fig. S-2. Room temperature photoluminescence of an exfoliated GaSb membrane, compared to a GaSb substrate.

PL measurements (Fig S-2): PL measurements of a 920 nm thick GaSb membrane (blue), show a similar peak position and fwhm as PL of a GaSb (001) substrate (black).

Figure S-3. Magnetotransport.

Magnetotransport measurements (Fig. S-3) show that films grown on graphene have similar carrier mobility and carrier concentration as films grown directly on GaSb (001) substrates. In Fig. S-3, the doped GaSb films on graphene/GaSb (001) and on bare GaSb (001) were grown simultaneously by co-loading the two substrates, in order to eliminate variations in growth conditions. The epitaxial structure consists of an unintentionally doped (uid) GaSb buffer layer [250 nm], Ge-doped GaSb [100 nm], and a uid GaSb cap [200 nm]. We use Ge as a dopant because our MBE system is not equipped with conventional Si or GaTe dopant sources.

Below the freeze-out temperature of native acceptors in the uid GaSb substrate (~60K), the two samples show similar electron concentration and electron mobility. These mobilities are similar to some reports of MBE-grown GaSb [R. Wiersma, et. al. PRB 67, 165202 (2003)], but lower than optimized GaSb quantum wells and modulation doped heterostructures. We attribute the moderate mobility in both samples to several factors: (1) the presence of carbon from graphene or organic residuals, which is a known acceptor in GaSb with low mobility of ~40 cm²/Vs at 10 K [R. Wiersma, et. al. PRB 67, 165202 (2003)], (2) amphoteric doping of Ge [E. Longenbach, et. al. J. Appl. Phys. 69, 3393 (1991)], (3) the high concentration of native acceptors in GaSb [G. Turner, et. al. JVST B, 11, 864 (1993)], (4) other transition metal

impurities from our MBE system. Regarding #4, note that our MBE system is also used for growth of intermetallic Heusler compounds such as Ni₂MnGa, FeVSb, and GdPtBi. Rare earth or transition metal impurities can be a source of deep level traps that decrease the carrier mobility, compared to films grown in a dedicated III-V MBE system.

Our measurements also suggest important questions about the role of carbon doping in other III-V materials grown on graphene, and interface scattering or screening from the graphene. Unfortunately we could not find Hall effect measurements reported for the previous III-V films grown by “remote epitaxy” [W. Kong, et. al. Nature Materials 17, 999 (2018), S-H Bae, et. al. Nature Nano 15, 272 (2020), Y. Kim, et. al. Nature 544, 340 (2017)]. Therefore we are not able to directly compare the transport characteristics of our GaSb samples with the previous reports of remote epitaxy. We agree with the reviewer that understanding the transport characteristics of films grown on graphene is crucial for a full understanding of potential device applications.

We also note that our transport measurements were performed on an unexfoliated sample that sits on the graphene/substrate. The use of metal stressor layers for exfoliation necessitates secondary processing steps to remove the metal stressor, which would be a parasitic conduction path, and placing it on a rigid mechanical support. As such, measurements on a heavily processed membrane may be a less reliable metric of the native quality of films grown on graphene. As a confirmation that our transport measurements are representative of a film grown on graphene, we emphasize that (1) measurements are below the freeze-out temperature of native acceptors, and thus transport occurs in the doped film (or graphene?), and (2) we were able to exfoliate this sample after the transport measurements were completed.

In addition, the growth mechanism is not still clear. If the remote epitaxy cannot be ruled out for the film growth as the authors mentioned, the authors' argument on the pin-hole seeded epitaxy and results could not be strong to be published in this journal.

Our results unambiguously demonstrate that GaSb films grow on graphene/GaSb(001) via a seeded lateral epitaxy mechanism, due to the high concentration of pinholes. Remote interactions are not the dominant mechanism. The major impact is that previously published reports on “remote epitaxy” may not provide sufficient proof of the intended remote mechanism, because they do not quantify the defects in graphene immediately prior to III-V film growth. These defects can seed lateral epitaxy. Our experiments establish a higher bar for experimental proof of “remote epitaxy.”

Our experiments replicate the same essential features of previous reports on remote epitaxy and show that the growth can be entirely explained by pinholes. We use the same graphene transfer procedure, and recent XPS measurements show a similar concentration of oxides at the graphene/substrate interface [H. Kim, et. al., ACS Nano, 15, 6 (2021)]. Our key finding is that desorption of these interfacial oxides creates pinholes, which seed selective nucleation and lateral growth. Previous reports have not quantified the graphene quality after this crucial native oxide desorption step, and thus the previous reports cannot rule out a pinhole mechanism. We show that any claim of a remote epitaxy mechanism needs to measure the defects in graphene, as they appear immediately before epitaxial film growth. Measuring the graphene quality immediately after transfer, but before pre-growth annealing, is not sufficient.

Clean graphene/substrate interfaces with defect spacing greater than the diffusion length ($L \gg \lambda$) are necessary in order to evaluate a possible remote epitaxy mechanism.

Changes to the text: At the end of II.D “Comparison with remote epitaxy” on page 8, we clarify this distinction between seeded lateral epitaxy and remote epitaxy, and provide an outlook on how remote epitaxy could be proven experimentally using lower defect density graphene.

Reviewer #2 (Remarks to the Author):

The authors implemented several changes in the revised version of the manuscript based on my (and the other reviewers) comments and criticism. Thus, it is my opinion that the manuscript has improved substantially and is now suitable for publication in Nature Communications.

We thank the reviewer.

Reviewer #3 (Remarks to the Author):

The authors have satisfactorily addressed my concerns and (I believe) those of the other reviewers as well. The paper is now much stronger than the initial submission and I believe it should be published in Nature Communications.

We thank the reviewer.

REVIEWER COMMENTS

Reviewer #1 (Remarks to the Author):

The revised manuscript offers the room temperature PL spectra and magnetotransport data of the samples. The room temperature PL spectra are quite broad so that they may not tell the difference between the samples. In addition, low temperature PL should be required to investigate impurity incorporation during the growth and accurately compare the PL characteristics of the samples. More importantly, for the magnetotransport measurement of the GaSb grown on graphene, the authors should consider the effect of the graphene on the transport characteristics because graphene between GaSb film and substrate could be conducting. So, without lifting off the sample from the substrate and removing the graphene, the transport measurement would be meaningless.

In addition, I think that the XRD data of the GaSb/graphene/GaSb cannot tell the crystallinity of the GaSb film if it is not removed from the substrate because the thick GaSb substrate would give the dominant signal for the XRD data in Fig. S-1. Overall, since the experiments to investigate the qualities of the samples are not performed carefully, I would not recommend to publish this paper to the Nature Communications.

Feb 20, 2022

We thank Reviewer 1 for their careful consideration. Please see below red text for our responses. There are no changes to the main text.

Reviewer #1 (Remarks to the Author):

The revised manuscript offers the room temperature PL spectra and magnetotransport data of the samples. The room temperature PL spectra are quite broad so that they may not tell the difference between the samples. In addition, low temperature PL should be required to investigate impurity incorporation during the growth and accurately compare the PL characteristics of the samples. More importantly, for the magnetotransport measurement of the GaSb grown on graphene, the authors should consider the effect of the graphene on the transport characteristics because graphene between GaSb film and substrate could be conducting. So, without lifting off the sample from the substrate and removing the graphene, the transport measurement would be meaningless.

In addition, I think that the XRD data of the GaSb/graphene/GaSb cannot tell the crystallinity of the GaSb film if it is not removed from the substrate because the thick GaSb substrate would give the dominant signal for the XRD data in Fig. S-1. Overall, since the experiments to investigate the qualities of the samples are not performed carefully, I would not recommend to publish this paper to the Nature Communications.

We appreciate the reviewer's interest in the electronic, optical, and structural properties of films grown on graphene. We agree that these properties are important for future device applications and for fundamental physics. However, a detailed optimization and analysis of film quality misses the main purpose of this paper.

The purpose of this paper is to show that a pinhole mechanism explains the epitaxy of GaSb films on graphene, producing films that are at least of equal materials quality as previous reports of remote epitaxy. We show how a seeded lateral epitaxy mechanism is strongly supported by the microscopic measurements (AFM, SEM, XPS, RHEED), using the same graphene transfer and similar film growth procedures as the original reports of remote epitaxy. Direct microscopic evidence for a remote epitaxy mechanism has not previously been reported, and has only been postulated based on the ability to exfoliate films and based on idealized DFT calculations in absence of defects. Our measurements show that the ability to exfoliate is not proof of growth by a remote mechanism, and that real samples produced by the published methods can have pinhole defects.

To emphasize the equivalent materials quality that places our claims of a pinhole mechanism on strong footing, in Table 1 below we compare the metrics for our GaSb films and exfoliated membranes with previous reports of remote epitaxy. In all metrics, our films

display equivalent or better quality as the previous reports. In many cases, e.g. transport, our measurements go beyond the previous studies and there are no available comparisons. We believe that a detailed analysis point and extended defects, doping from graphene, and the low temperature PL and transport scattering mechanisms are excellent topics for a follow up study. But these detailed studies are beyond the scope of the current paper, which focuses on mechanisms.

Regarding specific measurement requests:

1. Low temperature PL: We agree that a low temperature PL measurement may be more insightful than room temperature PL. However, this is beyond the scope of this paper. In fact, we could not find any previous reports of low temperature PL for films grown by remote epitaxy. To our knowledge, all PL in the remote epitaxy papers were performed at room temperature.
2. Transport of membrane: Similarly, we also agree that transport of a free-standing membrane is an important measurement to isolate possible doping effects of graphene. However, the Ni stressor used for exfoliation is a parallel conduction path, and it is challenging to remove this Ni layer from an ultrathin membrane without damaging the membrane. Therefore, such a measurement would require significant development of processing recipes. In the previous reports of remote epitaxy we were not able to find any transport measurements, either before or after exfoliation. Therefore, our report of transport pre-exfoliation is a new measurement that has never previously been reported.
3. X-ray diffraction of exfoliated membrane: Main text Fig 1g already contains a 2θ - 2θ XRD measurement of an exfoliated membrane, which exhibits a narrow fwhm of 0.05 degrees. This is the narrowest fwhm of any III-V membrane released from graphene, as shown in Table 1 below. The referee expresses concern about the interpretation of the rocking curve in Fig. S-1a (blue curve), since the rocking curve is for a film that has not been exfoliated. The fact that we observe a secondary peak shifted to lower ω allows us to identify this peak with fwhm = 12.6 arc seconds as coming from the film itself, and not the substrate.

metric	This work, GaSb	Y. Kim, Nature 544, 340-343 (2017), GaAs	W. Kong, Nature Materials 17, 999-1004 (2018), GaAs, GaN, LiF	J Jiang. Nature Communications 10, 4145 (2019), CsPbBr3	N. Wang. Applied Surface Science, 585, 152709 (2020) ZnO	H. Kum. Nature, 578, 75-81 (2020), SrTiO3, BaTiO3, CaFeO3, YIG	Y Guo, Nano Letters 20, 33-42 (2020), VO2	Z. Lu. Nanotechnology, 29, 445702 (2018), Cu	J. Yoo, Nanoscale 10, 5689-5694 (2018), Ge	Y. Alaskar, Advanced Functional Materials, 24, 6629-6638 (2014), GaAs
XRD rocking curve, omega fwhm	12.6 arcsec for film on graphene	not reported	not reported	1404 arcsec on graphene	4594 arcsec on graphene	not reported	180 (strained), 1700 (relaxed) arcsec on graphene	not reported	202 arcsec on graphene	245 arcsec on graphene
XRD peak width, 2theta fwhm	0.05° for released membrane	not reported	0.06 deg GaN membrane	0.29 deg film on graphene	1.6 deg film on graphene	not reported	0.27 deg film on graphene	0.125 deg film on graphene	not reported	0.25 deg film on graphene
Carrier mobility	285 cm ² /Vs at 2 K for film on graphene	not reported	not reported	not reported	not reported	not reported	not reported	not reported	not reported	not reported
Steady state PL linewidth, 300 K	56 meV released membrane, 62 meV substrate	45 meV membrane, 39 meV substrate	not reported	79 meV on graphene/NaCl, 81 meV on bare NaCl	not reported	not reported	not reported	not reported	not reported	not reported
RHEED	streaky c(2x6) reconstruction	not reported	not reported	not reported	not reported	streaky (1x1) SrTiO3	not reported	not reported	not reported	not reported
AFM	Atomic step morphology	Atomic step morphology	not reported	islands and nm step bunching	not reported	Atomic step morphology	island morphology (SEM)	island morphology	island morphology (SEM)	island morphology (SEM)

Table 1: Metrics for electronic, optical, and structural quality of films grown on graphene. Green shading indicates the highest quality sample for that particular metric (smallest fwhm or linewidth compared to reference sample, highest mobility, sharpest RHEED). Red shading indicates no measurement reported.

REVIEWER COMMENTS

Reviewer #2 (Remarks to the Author):

I have reviewed the latest version of the manuscript entitled "Pinhole-seeded lateral epitaxy and exfoliation on graphene-terminated surfaces" as well as the rebuttal letter of the authors which addresses the concerns raised by the last reviewer. With exception of the reviewer's comment on the magneto-transport results (see comment below), I agree with the authors that the reviewer's suggestions are certainly reasonable, but addressing all of the mentioned aspects goes beyond the scope of the manuscript. Thus, I believe that after the authors perform a minor revision based on the comment below, the paper is suitable for publication in Nature Communications.

- Although a 250 nm thick uid GaSb layer is placed between Ge:GaSb and graphene, it could be possible that pinholes in the uid GaSb layer would allow for transport to take place in the graphene layer as well. In other words, the sample with graphene would have two parallel transport channels, meaning that the final carrier concentration and mobility values will depend on the conductivity of both Ge:GaSb and graphene which cannot be easily disentangled. I suggest the authors include some discussion on that (considering this possibility or ruling it out in case they have experimental evidence for that). Ideally, they should show data for a similar vdP measurement performed on a bare graphene/GaSb template for a more complete comparison. Such measurement should not be time-consuming and would not require further work on material quality optimization, etc.

Reviewer #3 (Remarks to the Author):

I was favorably impressed by the earlier version of the manuscript and recommended publication in Nature Communications; the revised manuscript is further strengthened and I believe it should be published in its current form in Nature Communications. Below, I focus on the key questions brought up in the most recent round of reviews.

As the authors tactfully point out in their most recent revision letter, the suggestions from Reviewer #1 for further optimization and additional materials characterization simply miss the main purpose of this manuscript. Previous work on remote epitaxy has not demonstrated conclusively that remote epitaxy actually occurs, only that experimental results and DFT calculations are consistent with the hypothesis of remote epitaxy. By contrast, this paper conclusively demonstrates that pinholes can indeed be present and one still observe all the hallmarks of remote epitaxy (i.e. excellent quality grown materials that can be subsequently transferred). As is seen in Table I of the revision letter, it is clear that this manuscript presents the most thorough and systematic characterization to-date in the remote epitaxy field by a substantial margin.

To the specific characterization techniques discussed in the revision letter:

1. Low-temperature PL: PL is inherently a dicey metric whenever one must compare samples of differing structure. In this case, it is unavoidable that one ends up comparing apples with oranges and I find it highly unlikely that low-temperature PL would reveal anything about the fundamental band-to-band optical transition that room-temperature PL has not. Looking at Fig. S-2, one can see that both the signal-to-noise ratio (a good proxy for PL strength, even if the two curve heights have been normalized to the same peak height) and full-width-at-half-maximum (FWHM, a decent proxy for structural quality under the same excitation density) are comparable between the substrate and 'membrane' samples. It would be helpful if the authors indicated whether or not the curves in Fig. S-2 were normalized to one another. At any rate, to within the limits of PL, this suggests that the two are of comparable quality.

2. Transport on free-standing membrane: I agree that this would be ideal to eliminate any possible doping effects from graphene; however, I concur with the authors that the issues associated with removing the Ni layer without damaging the film make it fall outside the scope of this paper. Performing this experiment would only tell you how good the sample quality was after growth and processing, which is not necessarily the same thing as its quality right after epitaxial growth.

3. XRD of exfoliated membrane: I concur with approach that the authors took in analyzing the rocking curve measurements shown in Fig. S-1a to determine the FWHM of the grown film. Additionally, I agree that this FWHM is better than what has been reported in the remote epitaxy literature. Is it a little weird that there is a peak splitting? Yes, but similar effects have been observed in high-quality lateral epitaxial overgrown structures on patterned SiO₂ that are yet unexplained; I think exploring this effect falls outside the scope of this paper. Additionally, the pole figure results in Fig. S-1b provide further support that the structures grown on graphene are of comparable quality to those grown conventionally.

In summary, I do not think further characterization is necessary for this paper and the manuscript should be published in Nature Communications.

Reviewer 2

I have reviewed the latest version of the manuscript entitled “Pinhole-seeded lateral epitaxy and exfoliation on graphene-terminated surfaces” as well as the rebuttal letter of the authors which addresses the concerns raised by the last reviewer. With exception of the reviewer’s comment on the magneto-transport results (see comment below), I agree with the authors that the reviewer’s suggestions are certainly reasonable, but addressing all of the mentioned aspects goes beyond the scope of the manuscript. Thus, I believe that after the authors perform a minor revision based on the comment below, the paper is suitable for publication in Nature Communications.

- Although a 250 nm thick uid GaSb layer is placed between Ge:GaSb and graphene, it could be possible that pinholes in the uid GaSb layer would allow for transport to take place in the graphene layer as well. In other words, the sample with graphene would have two parallel transport channels, meaning that the final carrier concentration and mobility values will depend on the conductivity of both Ge:GaSb and graphene which cannot be easily disentangled. I suggest the authors include some discussion on that (considering this possibility or ruling it out in case they have experimental evidence for that). Ideally, they should show data for a similar vdP measurement performed on a bare graphene/GaSb template for a more complete comparison. Such measurement should not be time-consuming and would not require further work on material quality optimization, etc.

We thank the reviewer for this suggestion. Based on the suggested control measurement of the graphene/GaSb sheet resistance, it is difficult to rule out graphene as a parallel conduction path. Given the additional processing challenges of transport on an exfoliated sample, which may impart chemical or mechanical damage when removing the Ni stressor layer, we have decided to remove the transport measurements from this manuscript (previous figure S-3). This removal does not change the scientific message of the paper. We explain our findings below.

This figure compares the sheet resistance of our doped Ge:GaSb / graphene / GaSb heterostructure (blue curve) with the sheet resistance of graphene/GaSb alone (black curve). Both samples were measured in a van der Pauw geometry using annealed indium contacts. Before making contacts, the graphene/GaSb sample was annealed in UHV above the GaSb native oxide desorption temperature (530°C) to produce the same graphene pinhole morphology as in the full heterostructure. The sheet resistances of the two samples are within a factor of 10, making it difficult to rule out graphene as a parallel conduction path.

There are at least three possible scenarios to explain electrical contact to graphene. The first is graphene pinholes as the reviewer mentions, which may cause roughness in the overgrown GaSb films and direct contact between the Ge-doped GaSb layer and the underlying graphene. However, we note that there is a 250 nm undoped GaSb spacer layer in between, and our AFM and SEM measurements suggest that after only 60 nm (200 monolayers) of GaSb growth, coalescence is nearly complete with an atomically smooth film over a ~1 micron length scale (Fig 4 of the main manuscript). However, the 60 nm GaSb film on graphene does show larger holes, spaced a few microns apart. It is unclear whether similar large holes (although sparse) may appear for the 250 nm undoped GaSb spacer. A second possible source of contact is unintentional carbon doping of the GaSb spacer. We do not have direct measurements to rule this out. A third possibility is that the indium contacts may diffuse all the way down to the graphene itself. This in principle can be estimated from the diffusion coefficients of indium in GaSb; however, we don't have a direct measurement to rule it out. Since significant follow up experiments are required to rule out these scenarios, and since the transport properties of GaSb films are beyond the scope of the paper, we have decided to remove the transport measurements from the supplemental information.

Reviewer #3 (Remarks to the Author):

I was favorably impressed by the earlier version of the manuscript and recommended publication in Nature Communications; the revised manuscript is further strengthened and I believe it should be published in its current form in Nature Communications. Below, I focus on the key questions brought up in the most recent round of reviews.

As the authors tactfully point out in their most recent revision letter, the suggestions from Reviewer #1 for further optimization and additional materials characterization simply miss the main purpose of this manuscript. Previous work on remote epitaxy has not demonstrated conclusively that remote epitaxy actually occurs, only that experimental results and DFT calculations are consistent with the hypothesis of remote epitaxy. By contrast, this paper conclusively demonstrates that pinholes can indeed be present and one still observe all the hallmarks of remote epitaxy (i.e. excellent quality grown materials that can be subsequently transferred). As is seen in Table I of the revision letter, it is clear that this manuscript presents the most thorough and systematic characterization to-date in the remote epitaxy field by a substantial margin.

To the specific characterization techniques discussed in the revision letter:

1. Low-temperature PL: PL is inherently a dicey metric whenever one must compare samples of differing structure. In this case, it is unavoidable that one ends up comparing apples with oranges and I find it highly unlikely that low-temperature PL would reveal anything about the fundamental band-to-band optical transition that room-temperature PL has not. Looking at Fig. S-2, one can see that both the signal-to-noise ratio (a good proxy for PL strength, even if the two curve heights have been normalized to the same peak height) and full-width-at-half-maximum (FWHM, a decent proxy for structural quality under the same excitation density) are comparable between the substrate and 'membrane' samples. It would be helpful if the authors indicated whether or not the curves in Fig. S-2 were normalized to one another. At any rate, to within the limits of PL, this suggests that the two are of comparable quality.

Yes, the two PL spectra were normalized to one another. We have updated the caption in Fig. S-2.

2. Transport on free-standing membrane: I agree that this would be ideal to eliminate any possible doping effects from graphene; however, I concur with the authors that the issues associated with removing the Ni layer without damaging the film make it fall outside the scope of this paper. Performing this experiment would only tell you how good the sample quality was after growth and processing, which is not necessarily the same thing as its quality right after epitaxial growth.

3. XRD of exfoliated membrane: I concur with approach that the authors took in analyzing the rocking curve measurements shown in Fig. S-1a to determine the FWHM of the grown film. Additionally, I agree that this FWHM is better than what has been reported in the remote epitaxy literature. Is it a little weird that there is a peak splitting? Yes, but similar effects have been observed in high-quality lateral epitaxial overgrown structures on patterned SiO₂ that are yet unexplained; I think exploring this effect falls outside the scope of this paper. Additionally, the pole figure results in Fig. S-1b provide further support that the structures grown on graphene are of comparable quality to those grown conventionally.

In summary, I do not think further characterization is necessary for this paper and the manuscript should be published in Nature Communications.

REVIEWERS' COMMENTS

Reviewer #2 (Remarks to the Author):

In this revised version of the manuscript, the authors included additional changes suggested by reviewer #3. In addition, they decided to remove the transport results from the suppl. material, due to potential uncertainties associated with the measurement (which I pointed out in my previous revision). The authors argue that "this removal does not change the scientific message of the paper", which I agree. Thus, it is my opinion that the manuscript in its current form should be accepted for publication in Nature Communications.

Please wait...

If this message is not eventually replaced by the proper contents of the document, your PDF viewer may not be able to display this type of document.

You can upgrade to the latest version of Adobe Reader for Windows®, Mac, or Linux® by visiting http://www.adobe.com/go/reader_download.

For more assistance with Adobe Reader visit <http://www.adobe.com/go/acrreader>.

Windows is either a registered trademark or a trademark of Microsoft Corporation in the United States and/or other countries. Mac is a trademark of Apple Inc., registered in the United States and other countries. Linux is the registered trademark of Linus Torvalds in the U.S. and other countries.